# Mismatch between media coverage and research on invasive species: The case of wild boar (*Sus scrofa*) in Argentina

Sebastián A. Ballari[1]*, M. Noelia Barrios-García[1,2]

1 Parque Nacional Nahuel Huapi (CENAC-APN) and Consejo Nacional de Investigaciones Científicas y Técnicas (CONICET), San Carlos de Bariloche, Río Negro, Argentina, 2 Rubenstein School of Environment and Natural Resources, University of Vermont, Burlington, Vermont, United States of America

☺ These authors contributed equally to this work.
* sebastianballari@gmail.com

**Data Availability Statement:** All relevant data are within the paper and its Supporting Information files. The raw data to carry out this study was uploaded to the free access OSF Home repository: DOI 10.17605/OSF.IO/TDH4J (https://osf.io/tdh4j/).

## Abstract

Invasive species are a pervasive driver of global change with increasing media coverage. Media coverage and framing can influence both invasive species management and policies, as well as shed light on research needs. Using the wild boar (*Sus scrofa*) invasion in Argentina as a case study, we conducted a content analysis of media coverage and scientific articles. Specifically, we compared news and scientific articles based on their emphasis: ecological, economic, and health impacts and the overall perception portrayed in the news: "positive" when the articles emphasized benefits from wild boar and "negative" when focused on damage and/or loss. A literature search using Google news, Web of Science, Scielo, and Google Scholar yielded a total of 194 news articles and 37 research papers on wild boar in Argentina. More than half of the news articles focused on economic impacts of wild boar (56%) such as sport hunting, illegal hunting, and road accidents; while 27% focused on ecological impacts, and 10% on health impacts. In contrast, the majority of the scientific articles (65%) focused on ecological impacts of wild boar on native species and ecosystems; while 21% were related to health impacts and only 8.3% of scientific articles were related to economic impacts. This mismatch between media and science reveals a disconnection between social and scientific interests in wild boar and their management in Argentina, and it provides insights to research needs and prevention of management conflicts. Additionally, we found that 66.8% of news articles focused on "negative" aspects of wild boar, while 33.2% of news articles portrayed "positive" perceptions. This finding is very important because the management of invasive species such as wild boar usually requires lethal techniques, and the success of the programs depend on favorable social and political support. Good science communication is therefore key to helping scientists and managers perform more effective management actions.

**Funding:** The authors received no specific funding for this work.

**Competing interests:** The authors have declared that no competing interests exist.

## Introduction

Media can play a key role in forming opinions by influencing people's interests, understanding, perception of a specific topic, and potentially action, through news and/or social media [1–3]. Increasing amounts of media coverage in recent decades have accompanied public interest in environmental issues [4]. For example, a greater amount of climate change media coverage increased people's understanding of the environmental problem (e.g. [5–7]), which became one of the greatest challenges faced by humankind [8]. Similarly, high visibility in media with respect to a species coincides with a high level of perception of that species and vice versa [9]. This phenomenon can potentially "improve" the image of a species, or conversely "demonize" it, turning public opinion to a negative perception. For example, media coverage of negative events, such as predator attacks (e.g. mountain lion on livestock), can amplify perceived risk and reduce support for conservation interventions [10].

Successful conservation strategies depend on favorable social, political, and ecological conditions [10]. Therefore, the framing of media coverage is key to influencing public engagement, as well as government policy agendas, by increasing or giving prominence to a particular issue. Framing is the process through which the media selects certain aspects of an issue or event to emphasize [11]. Salience and coalescence of a topic is often translated into pressure on government officials to prioritize development of policy solutions [12]. For example, newspapers helped early conservationists sound the alarm over the disappearance of endangered bird species due to the popularity of feather plumes in women's hats [4]. Similarly in the 1960s, the media reported the spread of DDT pesticides through the food chain and its devastating effects on birds [4]. Likewise, media coverage can also influence invasive species management and policy.

Invasive species are a pervasive driver of global change with growing media representation (e.g. [12–15]). Previous studies have shown that the perception of invasive species can be influenced both positively or negatively by the way species are portrayed, via increased media exposure or by emphasizing specific points of view [16–18]. Similarly, invasive species management can become a contentious issue, as media can increase or decrease the discontent or social support of management strategies, which can subsequently determine the fate of a management plan (e.g. [17, 18]). For example, media coverage (i.e. news) and protesters changed the management strategy of the hedgehog, an introduced species in Scotland, from trapping and euthanasia to a translocation initiative [17]. In addition, other studies have shown that the number of news articles and framing of invasive species impacts can influence legislator and public support of invasive species control and prevention [13]. For example, Miller et al. [12] found that congressional policy activity in the United States increased with media news, highlighting the negative impacts of wild boar on agriculture. Because news media can help establish parameters for public discourse, or how people think and talk about a subject [19], understanding the nature of media coverage may help scientists and managers perform more effective management activities [10] and/or identify research needs.

The wild boar (*Sus scrofa*), also known as wild pig, feral pig, swine, or hog [20], is one of the most widely distributed invasive species throughout the world [21]. This species, native to Eurasia and North Africa, is considered one of the most harmful invasive alien species, endangering not only biodiversity but also the economy and human well-being [21, 22]. The wild boar is a generalist omnivore that feeds by overturning extensive areas of soil and vegetation. This disturbance causes many ecosystem-level effects by altering soil processes, reducing plant productivity, altering habitat availability, and threatening biodiversity conservation [21, 23–25]. In addition, wild boar disturbance and predation has an economic impact on human

productive systems as it is considered a crop pest, livestock predator, and competitor for forage and water resources, as well as a vector for several serious diseases that can affect wildlife, farms, domestic animals, and humans [21, 26–28]. Nonetheless, human perception of the wild boar is complex, because different stakeholders (e.g. agricultural producers, politicians, conservationists, hunters) perceive socio-economic values of this species differently (e.g. recreational hunting, tourism, cuisine, [29, 30]).

Here, we explore the relationship between news media and scientific research on invasive species and how they can inform one another. We use the wild boar invasion in Argentina as a case study to conduct a content analysis in media coverage and scientific articles. Analysis of media coverage can provide insights into public opinion, stakeholder perspectives, and their influence on conservation actions [31]. We classified and compared news and scientific articles based on their emphasis: ecological, economic, and health impacts; as well as the overall perception ("positive" or "negative") and discussed the implications for invasive species management.

## Materials and methods

To identify newspaper articles on wild boar in Argentina we used Google search engine with the search terms "*Sus scrofa*" OR "jabalí" OR "chancho jabalí" OR "chancho salvaje" OR "chancho silvestre" OR "cerdo salvaje" OR "cerdo cimarron" AND "site:ar" to delimit the search to Argentina. We filtered results to include only news articles by choosing the "News" tab in Google. We then reviewed each article of the 338 found to determine whether it was relevant to our search (i.e. news that directly covers some aspect of the wild boar in Argentina) and eliminated those that did not deal with wild boar or were redundant occurrences of the same article (i.e., the same news story replicated in several media). This search yielded 194 news items published from 2007 through 2020 (S1 Table).

We then conducted a literature search in Web of Science, Scielo, and Google Scholar for scientific articles using the search terms "*Sus scrofa*" OR "wild boar" OR "wild pig" OR "feral pig" OR "jabalí" OR "chancho jabalí" OR "chancho salvaje" OR "chancho silvestre" OR "cerdo salvaje" OR "cerdo cimarron", AND "Argentina", and also checked the references cited therein. We set the search for all type of scientific articles and reviewed the abstract of each one to eliminate those focusing on several invasive species or on general topics in invasion biology. After the screening, 37 scientific articles published between 2003 and 2020 were relevant to our study (S2 Table).

We classified each news and scientific article according to the main topic: 1) Economic (crop damage, livestock predation, sport hunting, poaching, animal husbandry, commercial exploitation, road accidents); 2) Ecological (culling–reduction of population for conservation purposes-, soil disturbance, predation, population growth); 3) Health (transmission of diseases to humans and other animals, attacks); and 4) others (e.g. sightings). We also recorded the perception of the news: "positive perception" meant the item emphasized benefits from wild boar such as sport hunting, commercial exploitation, husbandry (breeding for sport hunting or meat), and domestic pets, while we considered "negative perception" to be focus on impacts such as crop damage, livestock predation, diseases, culling, use of public areas, car collisions, ecological impact (e.g. soil damage, predation, competition for resources, disease transmission), population growth, and attacks. An article could be associated with more than one topic and/or perception. We also recorded meta-data including web address, news source, publication date, city, and province. We compared the frequency of topics in news and scientific articles, and number of news and scientific articles among different geographic regions using a chi-square test in JMP (SAS Institute Inc.).

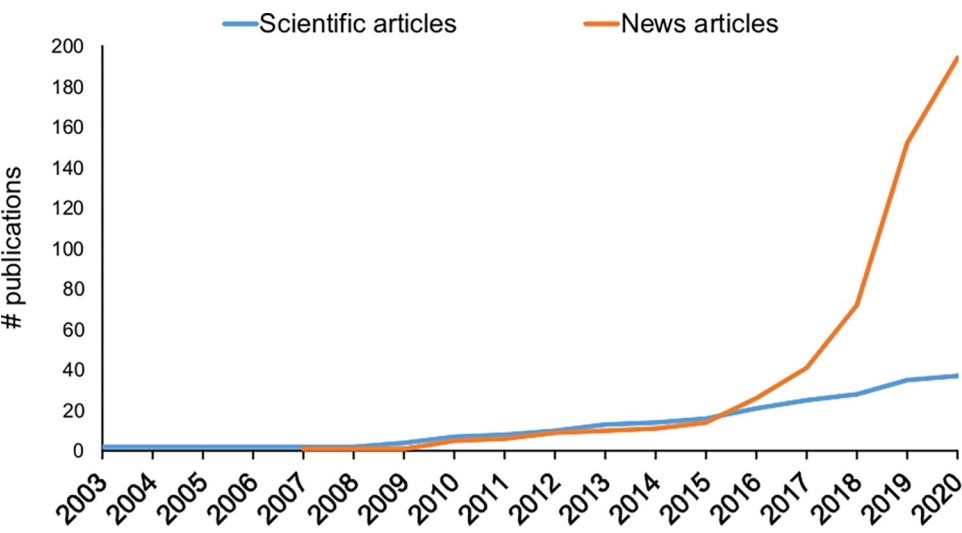

**Fig 1. Cumulative number of scientific and new articles on wild boar in Argentina.**

## Results

The first news article on wild boar in Argentina according to our search was published in 2007, then the number grew exponentially starting in 2017, reaching a total of 194 articles to date (Fig 1, S1 Table). In contrast, the first scientific article on wild boar in Argentina was published in 2003, and the number of publications grew slowly and steadily to 37 to date (Fig 1).

Topic frequency varied between news and scientific articles ($\chi^2$ = 38,847; d.f. = 3; p<0.0001). More than half of the news covered economic impacts of wild boar (56.1%), while ecological impacts reached 27% of the news, and health impacts almost 10% (Fig 2). Economic issues in the news covered several topics including sport hunting (12.6%), poaching (11.8%), road accidents (9.9%), commercial exploitation (8%), crop damage (6.1%), livestock predation (5.7%), and animals kept as pets (2%). Ecological issues in the news were related to population growth (11.4%), culling (10.7%), soil damage by rooting (3%) and competition or predation of native fauna (2%) (Fig 2). Finally, news on health focused on diseases (8%), and attacks on humans (1.5%) (S1 Table). By contrast, the majority of the scientific articles focused on

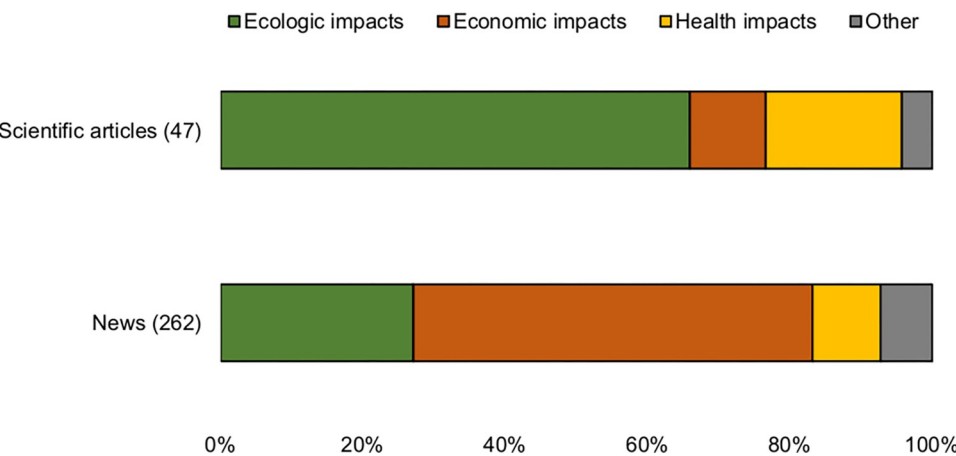

**Fig 2. Frequency of topics related to wild boar in scientific and new articles in Argentina until 2020.** An article could fit more than one topic.

ecological aspects (64.6%), 20.8% focused on health impacts, only the 8.3% of scientific articles are related on economic impacts. Ecological impacts (41.7%) includes references mainly based on wild boar rooting impacts (35.5%), and a few articles covered topics on population growth (12.5%) and culling (10.4%) (S2 Table). Most of the scientific articles on health impacts focused on diseases (18.7%), and only one article included information about attacks on humans (2.1%). Finally scientific articles on economic impacts focused on animal husbandry (4.1%), and road accidents and poaching (2.1% each) (Fig 2). Overall, we found that 66.8% of news articles focused on "negative" aspects of wild boar, while 33.2% of news articles portrayed "positive" perceptions about this species.

The number of news and scientific articles varied among geographic regions in Argentina ($\chi^2$ = 20.33; d.f. = 4; p = 0.0004). Most of the news articles (60%) were published in the Pampas region, which concentrates most of Argentina's agriculture (Buenos Aires, La Pampa, Cordoba, Santa Fe, and Entre Rios provinces, Fig 3). By contrast, scientific research on wild boar was mostly conducted in the Pampas and Patagonia regions (40% and 30%, respectively).

## Discussion

Our results show a mismatch on the focus of news and scientific articles published on wild boar in Argentina. News articles focused mainly on the economic impacts whereas scientific articles focused primarily on the ecological impacts of wild boar. These results likely reflect the interests of the reading audiences and provide insights in information gaps in the different reader communities. That is, media coverage tends to mirror the surrounding social environment and perceived interests of the readers [4], while scientific publications, which are generally not read by the public, are less influenced by social perception, and are thus somewhat dissociated from the social interests of the media (e.g., see Gozlan [9]).

Despite this disconnection, media coverage and scientific research can influence one another–and both scientific and socio-political activity–in important ways. The topics covered by the media can trigger new research, since media news showcases social, economic, or cultural aspects that generate massive interests and potentially reveal topics that require attention from the scientific community. An example is the prominence of economic impacts of wild boar in the media. Research on economic impacts of wild boar on human productive activities is scarce. Indeed, several studies show agriculture or plantation losses due to wild boar disturbance in the United States (e.g., [26, 32–34], but the lack of scientific information on economic losses plus difficulties in comparing different studies preclude the estimation of the annual economic impact [35]. This lack of information may also limit and/or hinder legislation and the design of management strategies. For example, the limitation or lack of knowledge of some aspects of invasive alien species has been recognized as one of the main threats to dealing with the management of these species in Europe [36].

Media coverage can influence public policies regionally or nationally. For example, Miller et al. [12] found that media coverage in the United States determines wild boar policy, as news on negative impacts resulted in an increased congressional policy activity and development. In Argentina, wild boar was declared a crop and livestock pest in the Pampas region where conflicts with agriculture arise (e.g. Decree 279. Resolution 272 of the Ministry of Health, Province of Buenos Aires, 2019). Wild boar legislation in this region allows year-round hunting to reduce wild boar damage in agricultural lands. However, there are no nation-wide bills like the United States' "Feral Swine Eradication and Control Pilot Program Act" or the "National Feral Swine Damage Management Program" [12].

Our results also show that while there are five times more news articles than scientific articles, the content on news articles does not reflect the scientific knowledge on wild boar

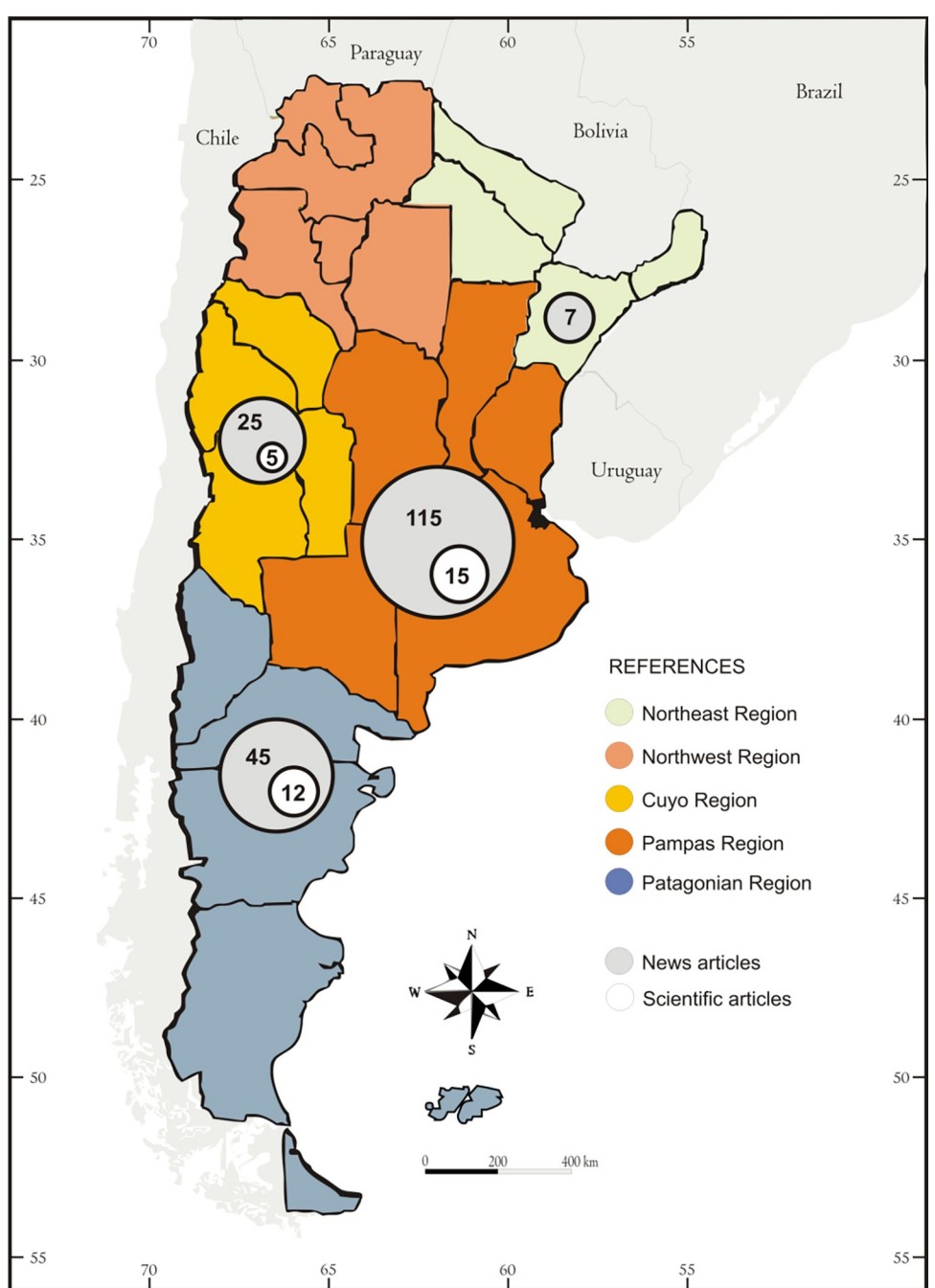

**Fig 3. Regional source of news and articles published on wild boar in Argentina until the year 2020.** The map was created based on the geographical classification of the Ministry of Environment and Sustainable Development of Argentina (https://www.argentina.gob.ar/ambiente).

invasion. The majority of the research on wild boar in Argentina was on ecological impacts, but this topic represented only 27% of the published news articles. Previous studies have shown that non-native species receive little media coverage [9, 15]; however, our results suggest there is considerable media attention on wild boar, but the aspects described often relate to social outcomes. This finding also suggests that scientists must better communicate their

knowledge to the public. Science communication is needed not only for increasing scientific literacy but also to avoid management conflicts [10, 13, 17] and ensure that management is evidenced-based [37].

Media can influence public perception and thus invasive species management decisions [10, 38]. Interestingly we found that "negative" perceptions were associated with destructive, damaging activities by wild boar, while "positive" perceptions came from economic benefits. This finding is in line with those by Rodriguez-Rey et al. [39] in which the popularity index of invasive species was consistent with the public perceptions of their ecological and economic damage. This scenario can be very important for the management of invasive species that are charismatic or captivating to the public (e.g. squirrels, deer, cats). For example, lethal control initiatives of a population of feral hippopotamuses (*Hippopotamus amphibius*) in Colombia were abandoned due to strong public opposition [18]. Hippopotamuses are a charismatic species and are valued by local communities because they attract tourists to the area [18]. Similarly, previous studies have found that the public's risk perception for invasive species influences their willingness to take action [40, 41]. Indeed, Vaske et al. [41] found that the acceptability of lethal management of wild pigs in the United States is related to people's beliefs and perception of risk. Similarly, a study conducted in Barcelona, Spain showed that 37% of the citizens interviewed answered that measures should be taken to minimize wild boar incidents [42]. For wild boar, there seems to be a consensus among news and scientific articles in which wild boar are depicted as problematic and dangerous, and that they must be controlled. This fact undoubtedly constitutes a key element for relevant authorities to undertake management and control plans across invaded ranges.

The topics of wild boar news and scientific articles in Argentina most likely reflect public interests worldwide. Both ecological and economic impacts are generally the most reported impacts of wild boar, providing the basis for different management strategies around the world (e.g., [43, 44]). It is somewhat surprising that health impacts were not prominent in either article type given the diseases and parasites that wild boar host and the potential impacts on both wildlife and human health [45], especially after several major outbreaks of the African Swine Fever worldwide (e.g., [46]). However, health impacts of wild boar will likely become a more conspicuous topic both in news and scientific articles after the covid-19 pandemic and consequently increased awareness of the importance of zoonotic diseases [47].

The news and research articles originated in the same regions where wild boar is distributed in Argentina. Currently, wild boar occupy a large part of Argentina, from northern Patagonia through central Argentina to the northeast [48]. However, most news on wild boar originated in central Argentina (the Pampas region), where the largest agricultural and livestock activity in the country is concentrated. Interestingly, only one of these articles focused on losses and damage in production of an agricultural crop. Instead, news from this area tends to focus on sport and illegal hunting. By contrast, scientific articles were based mainly on data from the Pampas and Patagonia regions, which matches the areas with longer history of presence and invasion of wild boar in Argentina [48, 49].

It is worth mentioning the increasing role of social media on invasive species research and management [3]. Social media, like other forms of communication, can be used effectively to share information about invasive species [50]. For example, Allain [51] found that the total number of a non-native freshwater turtle sightings within the UK recorded on Flickr from 2008 to 2018 was significantly greater than those submitted to Record Pool in the same period. Similarly, a crowdsourcing project on iNaturalist increased detection of wild boar (*Sus scrofa*) in Ontario, Canada [52]. Nonetheless, in the latter study the authors found that the reports of wild pig sightings were related to whether wild pigs were featured in media stories that directed the public on how to report sightings. This suggests that the number of reported wild

pig sightings in Ontario was not a direct index of the abundance of wild pigs on the landscape, but rather an artifact of time of year and our media and outreach efforts. While the relationship of social media and invasive species was not the aim of this study, future research should look into how social media can improve invasive species communication, research and management.

## Conclusion

A disparity characterizes the focus of the media and scientific articles. This mismatch potentially highlights information gaps in both types of articles and provides insights that can aid in preventing management conflicts. Future work should look into expanding the focus of media articles on ecological topics, while scientific articles should explore the economic impacts of wild boar. Media news and social perceptions are intimately connected and linked (e.g. [15]). In addition, when seeking a convincing story, media producers can contribute to the polarization of conservation issues by presenting two sides of a problem where one must choose "for or against" positions [17]. This polarization can lead to destructive conflicts and failed management actions. Therefore, increasing people's ecological awareness requires clear communication of ecological principles [53].

Invasive species management provides a good example of the need for good communication and conservation efforts: the management of an invasive species cannot be based solely on biological arguments and should be combined with social and economic considerations [54]. In fact, Lioy et al. [38] argued that media monitoring should be routinely included in the development of projects to manage biodiversity in order to evaluate the communication effectiveness and to help identify negative trends, thus facilitating proactive responses. Holistic socio-ecological approaches, where communication by media, science, and policy go hand in hand, are key to prevent, minimize, or mitigate the impacts of invasive non-native species.

## Supporting information

**S1 Table. Links to the news websites used in this study about wild boar in Argentina (N = 194).**
(DOCX)

**S2 Table. Scientific articles published on wild boar in Argentina (N = 37).**
(DOCX)

## Acknowledgments

We thank Dr. Daniel Simberloff, Dr. Christopher B. Anderson and Carolyn Hanrahan for reviewing earlier versions of the manuscript.

## Author Contributions

**Conceptualization:** Sebastián A. Ballari, M. Noelia Barrios-García.

**Data curation:** Sebastián A. Ballari, M. Noelia Barrios-García.

**Formal analysis:** Sebastián A. Ballari, M. Noelia Barrios-García.

**Investigation:** Sebastián A. Ballari, M. Noelia Barrios-García.

**Methodology:** Sebastián A. Ballari, M. Noelia Barrios-García.

**Project administration:** Sebastián A. Ballari, M. Noelia Barrios-García.

**Resources:** Sebastián A. Ballari, M. Noelia Barrios-García.

**Software:** Sebastián A. Ballari, M. Noelia Barrios-García.

**Supervision:** Sebastián A. Ballari, M. Noelia Barrios-García.

**Validation:** Sebastián A. Ballari, M. Noelia Barrios-García.

**Visualization:** Sebastián A. Ballari, M. Noelia Barrios-García.

**Writing – original draft:** Sebastián A. Ballari, M. Noelia Barrios-García.

**Writing – review & editing:** Sebastián A. Ballari, M. Noelia Barrios-García.

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
