## [Decision Letter · Decision Letter 0]

22 Aug 2022

PONE-D-22-17828Mismatch between media coverage and research on invasive species: the case of wild boar in ArgentinaPLOS ONE

Dear Dr. Ballari,

Thank you for submitting your manuscript to PLOS ONE. After careful consideration, we feel that it has merit but does not fully meet PLOS ONE’s publication criteria as it currently stands. Therefore, we invite you to submit a revised version of the manuscript that addresses the points raised during the review process.

We look forward to receiving your revised manuscript.

Kind regards,

Jorge Ramón López-Olvera

Academic Editor

PLOS ONE

Journal Requirements:

3. We note that Figure 3 in your submission contain map image which may be copyrighted. All PLOS content is published under the Creative Commons Attribution License (CC BY 4.0), which means that the manuscript, images, and Supporting Information files will be freely available online, and any third party is permitted to access, download, copy, distribute, and use these materials in any way, even commercially, with proper attribution. For these reasons, we cannot publish previously copyrighted maps or satellite images created using proprietary data, such as Google software (Google Maps, Street View, and Earth). For more information, see our copyright guidelines: http://journals.plos.org/plosone/s/licenses-and-copyright.

 a. You may seek permission from the original copyright holder of Figure 3 to publish the content specifically under the CC BY 4.0 license. 

Additional Editor Comments:

Dear Dr. Ballari,

thank you for considering PLoS ONE as the target journal to publish your research. I have now received the assessment of two independent reviewers on your submission and, as you will see, while both find the topic interesting and your manuscript potentially worth publication, they also point out some concerns (minor from one of the reviewers and major from the other) which should be addressed in order to improve your study before deserving publication.

The most relevant suggestion is to include some reference or even add a part of the study regarding social media, whose importance cannot be dismissed. I also suggest you to include some reference to the way the information published in media (both written, audiovisual, through internet and in social networks) influence the perception of wildlife, native and invasive species, in the line of the publication below:

Conejero C, Castillo-Contreras R, González-Crespo C, Serrano E, Mentaberre G, Lavín S, López-Olvera JR. 2019. Past experiences drive citizen perception of wild boar in urban areas. Mammalian Biology 96, 68–72. https://doi.org/10.1016/j.mambio.2019.04.002

I hope you find the comments from the reviewers useful in improving your manuscript. I am looking forward to receive the revised versions.

Best luck when carrying out your review.

Reviewers' comments:

Reviewer's Responses to Questions

**Comments to the Author**

1. Is the manuscript technically sound, and do the data support the conclusions?

Reviewer #1: Yes

Reviewer #2: Yes

2. Has the statistical analysis been performed appropriately and rigorously? 

Reviewer #1: Yes

Reviewer #2: Yes

3. Have the authors made all data underlying the findings in their manuscript fully available?

Reviewer #1: Yes

Reviewer #2: Yes

4. Is the manuscript presented in an intelligible fashion and written in standard English?

Reviewer #1: Yes

Reviewer #2: Yes

5. Review Comments to the Author

Reviewer #1: Your study is relevant and timely, especially since the literature on invasive species, and especially connecting news media and science community is still growing. Your manuscript is well organized, and clear, with each section clearly presented. Some more details could be given in the method section in terms of data analysis. Moreover, there is some newer literature that should me mentioned - for example the articles from Perry et al., 2022 in Diversity journal and from Perry et al., 2020. as well as Solano et al., 2022 and Koen et al., 2021 in Biological Invasions.

Reviewer #2: This is an interesting article that will certainly be useful for the literature upon careful revisions. I have listed line-by-line suggestions below. My two biggest qualms are as follows: (1) you are truly missing an opportunity to at least briefly mention social media. There is much work that discusses the use of social media in supporting conservation funding and removal efforts, and it’s also important to note that in some cases social media can instead support the spread of invasive species. Regardless, it’s been a large factor in many invasive species management strategies and it was surprising to see nothing listed about social media; (2) there is no clear portion of the manuscript that details “what’s next” – I’d like to see clear recommendations for what you think will be the best way to minimize the invasive wild boar invasion in Argentina to protect native wildlife. Thank you for the opportunity to read this interesting work.

Title, keywords, abstract

- The word “news” is in your abstract many times so I recommend a different word; perhaps “human-wildlife interaction” or a different relevant word that you didn’t include in your title or abstract

- Line 37: please include the database used for your literature search; please still with the phrase “scientific articles” or can you detail if these are primary academic studies

- Please include the species name for wild boar in the title or abstract

- Line 39: How are “road accidents” and “illegal hunting” economy related? Does Argentina restrict hunting of the invasive species? I wonder if these could be different categorized as “human-wildlife interactions”? What do the other 44% of articles discuss?

- Line 40: are these negative ecological impacts mostly? If there is space for more words, would be useful to know if scientific literature is mostly discussing the negative impacts (and not possible positive ones, like hunting opportunities etc. )

- The abstract is a bit unclear – are you suggesting that a match between what scientists are concerned about needs to match that of news media coverage, because the news can drive management efforts of wild boar? E.g., if people want to hunt for boar and the government culls the population, people would be upset and not understand they’re negative? Please try to rephrase the abstract to be as targeted with your goal and implications.

Introduction

- Line 60: please change “news or social media” to “news and/or social media”

- Line 61: What do you means by “increasing amounts of media”? People were interested in environmental issues regardless of media – do you mean that media coverage, especially with the growing massive use of social media, has been responsible for contributing to message sharing and awareness about environmental issues?

- Lines 62-64: media coverage of a topic doesn’t always convinced people it’s true!

- Lines 64-65: consider rephrasing this sentence to be more related to awareness e.g., “high visibility of species-specific media coverage can coincide with an increased level of perception and awareness of that species and its effects on the (local?) environment” – or something like that.

- Lines 65-67: this would entirely be based on the perspective from which that species is discussed, which you don’t mention – framing a species from a positive or negative perspective can result in the public perceiving a species as beneficial or harmful, respectively

- Line 69: “conservation” needs to be followed up with terms like management, strategies, actions, etc.

- Line 78: although your article, thus far, appears to focus on convention news outlets, Bergman et al. 2022 (https://doi.org/10.1139/facets-2021-0112) highlight many examples of the effects of social media on wildlife conservation & invasive species management efforts from much more recent dates – would be useful to highlight this as social media is now an inherent part of news coverage for many people

- Lines 80-81: I’m not sure what you mean here – media portrayals increased exposure? Not sure what these sentences mean.

- Line 83: why did you use two different types of quotation marks here? This sentence also seems contradictory – it’s set up to discuss contentious issues, I believe you’re attempting to say however that media can explain reasons that culling efforts are required and that can smooth opposition? Unclear

- Line 85: “social” and “media coverage” – is this supposed to be social media coverage? Or just news?

- Line 86: this example needs to be explained – don’t make us readers go searching for reference “13” – tell us what happened!

- Line 87-89: did media reach policy makers? Was it not science? Or did they need the public to get on board with management efforts? Unfortunately these examples don’t support your work well because you fail to elaborate on them and make the connections clear. Please pick one example and delve into it a bit more. Same with lines 88-90.

- Line 91: consider changing “US” to “the United States.” Also, the newspapers themselves are not “negative” but I assume you mean that they highlight the negative impacts of invasive wild boars on agricultural practices. Please consider rewording to make this clear.

- Line 91-92: are these all the same wild pig species? You need to include species names please.

- Line 92-95: you haven’t provided any real clear evidence of this yet so this sentence falls a bit flat and doesn’t convince readers

- Line 96: the species name should be included at the first mention of that species

- Lines 99-105: Consider rewriting this section to follow up each negative impact with an example e.g. you state they incur negative economic impacts, please then immediately follow up with your examples

- Line 106: what are the different stakeholder groups?

- Line 108-109: You frame the purpose of your article to include a management aspect, but then don’t include that in this sentence. Also, are you exploring “public interest and news media” AND “scientific research” as two separate components? This is how it reads.

- Line 109: insert “the” before “wild boar invasion.” Is the invasion across the entire country? Or localized to certain areas?

Materials and Methods

- Lines 118-119: were all of these terms in parenthesis?

- Line 121: how many articles in total did you review? How did you “sort” the results? By year or relevance?

- Line 123: consider change the “–“ to (i.e., the same news story replicated in several media)

- Line 125: I believe you should indent this new paragraph

- Lines 125-129: did you conduct a literature search in each of these databases separately using the exact same search terms? Did you use parenthesis? What language(s) did you include? Did you include review articles or only primary academic literature? How did you decide on those search terms? Why the year 1970?

- Line 135: this is inconsistent terminology – you used “ecological” before but now “ecologic”? What do you mean by culling – would this be management actions that are standardized and carries out to cull a large portion of a population but falls outside the class of consumption/hunting?

- Line 136: consider changing “health to “public health”

- Line 128: how would wild boar be related to “animal husbandry”? Consider changing “when kept as pets” to “suitable as domestic house pets” to make that portion of the sentence more clear

- Line 141: I believe you should change “topic and perception” to “topic and/or perception”

- Line 142: this is known as meta-data, please as a comma after city

- Line 144: how did you decide which geographic areas were “different”? Was it political or ecologically set?

- Line 147: be careful with your wording – this was the first article that your search revealed (our search strategies are never perfect, you may have missed something!)

- Line 151: ident the paragraph. You could refine wording and change to, “Topic frequency varied…”

- Line 153: the % doesn’t make sense here – how could ecological impacts “dominate” when it was less than 1/3? Do publications = scientific articles? Please keep terminology the same throughout the article if this is what you mean. Please change “barely” to uncommon or a word more similar to that.

- Line 156: can you give an example of ecological impacts? Perhaps a table that lists each of the categories within economic, ecological, and health would make it clearest for readers. Please add a comma after “finally”

- Lines 153-157: I don’t think your percentages are adding up, can you make clear which aspect falls under each category?

- Line 158: the 64.6% should be placed after “ecological aspects” and make this the end of a short sentence, starting the next sentence with your percentage information

- Line 159: “cover” should be “covered”

- Line 162: again, please place a comma after “finally” – 8.3% of what? Did you miss words here?

- Line 164-165: I think it might be useful to redo this aspect of your article – it’s not that the articles focused on negative aspects of wild boar (although they are indeed negative consequences they incur) but instead almost the media was accurately portraying the invasive species as invasive (which inherently means negative/destructive/damaging/ etc) whereas “positive” instead treats invasive wild boar as naturalized and a species that can offer economic pros. This is an aspect I don’t believe you’ve really discussed and is critical.

- Line 165-168: this is a method as you’ve already categorized and explained this

- Line 171: are the Santa Fe and Entre Rios the same area, or should you have added in a comma after Santa Fe?

- Line 172: add a % after 40, and a comma after 30%

Discussion

- Line 175: I don’t think the word “emphasis” is appropriate, it seems more to be the actual topic of focus that there is a mismatch between the media and academic worlds

- Line 176: remove the comma after “whereas”

- Line 181: choose a different word at least once instead of writing generally and general twice

- Line 182: why is this the first time you have a written-out citation? Is this a mistake?

- Line 183: I don’t believe this is striking…in fact, I’d say this is very common worldwide and what I’d expect

- Line 189-192: this is a long, somewhat confusing, sentence

- Line 192-193: how? Big statement but I don’t clearly see the connection…

- Line 195: this sentence needs to be rephrased to specifically describe that it was related to boar and not broadly all invasive species management

- Line 199: please change to “Invasive species legislation in the Pampas region allows for an extended hunting season and increase (??) area to control damages wild boar can incur to agricultural areas” or something like that to be more clear

- Line 200: please put the United States before those bills instead of at the end

- Line 206: please rephrase to, “; our results suggest there is considerable media attention focused on wild boar, however the aspects described often relate to social outcomes” or something

- Line 208: suggest needs to be plural; change “need to” to “must” and I would argue that this needs to be done in both conventional news outlets and also social media

- Line 210: not only to avoid management conflicts, but ensure that management is evidence-based and can support healthy wildlife populations and nature conservation

- Line 213-216: this is a long sentence and should be divided to give ample time to discussion about ecotourism and charismatic species (I’ll note however that this doesn’t seem to relevant to your article…)

- Line 222: inducement? This word doesn’t make sense here

- Line 224-225: you just said earlier though that the US is so different?

- Line 228: add “that” after “parasites”

- Line 229: Scientists traced the genetic lineage of H1N1 swine flu to a strain that emerged in 1998 in U.S. factory farms…swine flu is also emerging from wild boar? I didn’t know this…am I misinterpreting? The article you cite I believe refers specifically to ASF

- Line 230-232: consider selecting a word other than “probably” for this sentence and make the link clearer here between increased awareness about zoonotic diseases

- Line 234: did you list a statistical statistic for this finding? I may have missed that you found no significant difference between geographic areas and sources of news and research articles

- Line 234: remove the comma after Patagonia

- Line 236: put “the Pampas region” in parenthesis if this area is in central Argentina

- Line 238: production of an agricultural crop?

Figures

Love the map figure

6. PLOS authors have the option to publish the peer review history of their article (what does this mean?). If published, this will include your full peer review and any attached files.

Reviewer #1: No

Reviewer #2: No

---

## [Author Response · Author response to Decision Letter 0]

8 Nov 2022

PONE-D-22-17828

Mismatch between media coverage and research on invasive species: the case of wild boar in Argentina

Reply (RE):

Dear Dr. Ballari,

thank you for considering PLoS ONE as the target journal to publish your research. I have now received the assessment of two independent reviewers on your submission and, as you will see, while both find the topic interesting and your manuscript potentially worth publication, they also point out some concerns (minor from one of the reviewers and major from the other) which should be addressed in order to improve your study before deserving publication.

The most relevant suggestion is to include some reference or even add a part of the study regarding social media, whose importance cannot be dismissed. I also suggest you to include some reference to the way the information published in media (both written, audiovisual, through internet and in social networks) influence the perception of wildlife, native and invasive species, in the line of the publication below:

Conejero C, Castillo-Contreras R, González-Crespo C, Serrano E, Mentaberre G, Lavín S, López-Olvera JR. 2019. Past experiences drive citizen perception of wild boar in urban areas. Mammalian Biology 96, 68–72. https://doi.org/10.1016/j.mambio.2019.04.002

I hope you find the comments from the reviewers useful in improving your manuscript. I am looking forward to receive the revised versions.

Best luck when carrying out your review.

Response to the Editor:

Thank you for considering a revised version of our manuscript, please see our responses to the suggestions below. 

We appreciate the suggestion about social networks. In fact, it was something we considered in early stages of the study. However, we decided not to include them because the methodology for data collection is very different and because the sources we used (news and scientific articles) have some kind of review process (e.g. editorial editors, content editors, press officers, peer, associate editors, editors-in-chief). Additionally, while social networks represent a great source of information, it is mostly opinion and first-hand perception on topics that each user deems appropriate according to their interests. Therefore, the comparison of topics in social media to news or scientific articles seemed not quite appropriate in this case. However, we believe it is something to consider in future studies. Without any doubt, it is important to mention it and thus we have included the importance of social media in the discussion of the manuscript including some of the refences suggested, Lines: 257-269 reads: “It is worth mentioning the increasing role of social media on invasive species research and management [3]. Social media, like other forms of communication, can be used effectively to share information about invasive species [51]. For example, Allain [52] found that the total number of a non-native freshwater turtle sightings within the UK recorded on Flickr from 2008 to 2018 was significantly greater than those submitted to Record Pool in the same period. Similarly, a crowdsourcing project on iNaturalist increased detection of wild boar (Sus scrofa) in Ontario, Canada [53]. Nonetheless, in the latter study the authors found that the reports of wild pig sightings were related to whether wild pigs were featured in media stories that directed the public on how to report sightings. This suggests that the number of reported wild pig sightings in Ontario was not a direct index of the abundance of wild pigs on the landscape, but rather an artifact of time of year and our media and outreach efforts. While the relationship of social media and invasive species was not the aim of this study, future research should look into how social media can improve invasive species communication, research and management.”

Additional comments:

• Regarding the comments on the map (Figure 3), it was made by the authors using the Corel Draw program. The map is an original drawing that was created from scratch, without using any template or figure with copyright.

• The raw data to carry out this study was uploaded to the free access OSF Home repository: DOI 10.17605/OSF.IO/TDH4J (https://osf.io/tdh4j/)

 

Reviewers' comments:

Reviewer's Responses to Questions

Comments to the Author

1. Is the manuscript technically sound, and do the data support the conclusions?

Reviewer #1: Yes

Reviewer #2: Yes

2. Has the statistical analysis been performed appropriately and rigorously?

Reviewer #1: Yes

Reviewer #2: Yes

3. Have the authors made all data underlying the findings in their manuscript fully available?

Reviewer #1: Yes

Reviewer #2: Yes

4. Is the manuscript presented in an intelligible fashion and written in standard English?

Reviewer #1: Yes

Reviewer #2: Yes

5. Review Comments to the Author

Reviewer #1: Your study is relevant and timely, especially since the literature on invasive species, and especially connecting news media and science community is still growing. Your manuscript is well organized, and clear, with each section clearly presented. Some more details could be given in the method section in terms of data analysis. Moreover, there is some newer literature that should me mentioned - for example the articles from Perry et al., 2022 in Diversity journal and from Perry et al., 2020. as well as Solano et al., 2022 and Koen et al., 2021 in Biological Invasions.

Re: Thank you for your appraisal. We have added more details in the data analysis in Lines 134-148. Now reads: We classified each news and scientific article 

according to the main topic: 1) Economic (crop damage, livestock predation, sport hunting, poaching, animal husbandry, commercial exploitation, road accidents); 2) Ecological (culling – reduction of population for conservation purposes-, soil disturbance, predation, population growth); 3) Health (transmission of diseases to humans and other animals, attacks); and 4) others (e.g. sightings). We also recorded the perception of the news: "positive perception" meant the item emphasized benefits from wild boar such as sport hunting, commercial exploitation, husbandry (breeding for sport hunting or meat), and domestic pets, while we considered "negative perception" to be focus on impacts such as crop damage, livestock predation, diseases, culling, use of public areas, car collisions, ecological impact (e.g. soil damage, predation, competition for resources, disease transmission), population growth, and attacks. An article could be associated with more than one topic and/or perception. We also recorded meta-data including web address, news source, publication date, city, and province. We compared the frequency of topics in news and scientific articles, and number of news and scientific articles among different geographic regions using a chi-square test in JMP (SAS Institute Inc.). 

Additionally, we have added Koen’s reference suggested in Lines 262-263. We did not include Solano et al. because it did not seem relevant to our manuscript, and we did not include Perry et al 2020, 2022 because without the full citation we could not find any paper that relate to our study. 

 

Reviewer #2: This is an interesting article that will certainly be useful for the literature upon careful revisions. I have listed line-by-line suggestions below. My two biggest qualms are as follows: (1) you are truly missing an opportunity to at least briefly mention social media. There is much work that discusses the use of social media in supporting conservation funding and removal efforts, and it’s also important to note that in some cases social media can instead support the spread of invasive species. Regardless, it’s been a large factor in many invasive species management strategies and it was surprising to see nothing listed about social media; (2) there is no clear portion of the manuscript that details “what’s next” – I’d like to see clear recommendations for what you think will be the best way to minimize the invasive wild boar invasion in Argentina to protect native wildlife. Thank you for the opportunity to read this interesting work.

Re: Thanks for pointing out that we should mention social media. It was something we considered in early stages of the study, but because data collection is very different, and because the sources we used (news and scientific articles) have some kind of review process (e.g. editorial editors, content editors, press officers, peer, associate editors, editors-in-chief), we decided not to include it. Nonetheless, we added a paragraph on the importance of social media in the discussion section and suggested it as a future work, Lines 257-269: “: It is worth mentioning the increasing role of social media on invasive species research and management [3]. Social media, like other forms of communication, can be used effectively to share information about invasive species [51]. For example, Allain [52] found that the total number of a non-native freshwater turtle sightings within the UK recorded on Flickr from 2008 to 2018 was significantly greater than those submitted to Record Pool in the same period. Similarly, a crowdsourcing project on iNaturalist increased detection of wild boar (Sus scrofa) in Ontario, Canada [53]. Nonetheless, in the latter study the authors found that the reports of wild pig sightings were related to whether wild pigs were featured in media stories that directed the public on how to report sightings. This suggests that the number of reported wild pig sightings in Ontario was not a direct index of the abundance of wild pigs on the landscape, but rather an artifact of time of year and our media and outreach efforts. While the relationship of social media and invasive species was not the aim of this study, future research should look into how social media can improve invasive species communication, research and management. 

We appreciate your suggestion on “what’s next”, and we added in the conclusion (Lines 274-276): “Future work should look into expanding the focus of media articles on ecological topics, while scientific articles should explore the economic impacts of wild boar.”

Title, keywords, abstract

- The word “news” is in your abstract many times so I recommend a different word; perhaps “human-wildlife interaction” or a different relevant word that you didn’t include in your title or abstract

Re: We agree, we added “human-wildlife interaction” in the keywords.

- Line 37: please include the database used for your literature search; please still with the phrase “scientific articles” or can you detail if these are primary academic studies

RE: We included the databases for review in Lines 28-30: "A literature search using Google news, Web of Science, Scielo, and Google Scholar yielded a total of 194 news articles and 37 research papers on wild boar in Argentina.”

- Please include the species name for wild boar in the title or abstract

RE: Done

- Line 39: How are “road accidents” and “illegal hunting” economy related? Does Argentina restrict hunting of the invasive species? I wonder if these could be different categorized as “human-wildlife interactions”? What do the other 44% of articles discuss?

RE: Road accidents and illegal hunting imply an economic cost or /benefit to people or society. Road accidents incur a cost for the people who are injured or break their vehicles, as well as a cost for the cities that must remove the vehicles and repair the damage. On the other hand, illegal hunting implies an economic benefit for the hunters as they obtain meat that can be consumed or sold. Additionally, there is a cost for the governments to control illegal hunting activities. We added this information in Lines 30-35: “More than half of the news articles focused on economic impacts of wild boar (56%) such as sport hunting, illegal hunting, and road accidents; while 27% focused on ecological impacts, and 10% on health impacts. In contrast, the majority of the scientific articles (65%) focused on ecological impacts of wild boar on native species and ecosystems; while 21% were related to health impacts and only 8.3% of scientific articles were related to economic impacts.” 

Yes, Argentina still limits or restricts the hunting of this species in some regions (provinces). 

Since boar and wild pigs have a wide spectrum of impact at multiple levels, we believe it is convenient to categorize these impacts into a few categories. We selected the main categories used in our own previous studies (e.g. Barrios-Garcia and Ballari 2012).

- Line 40: are these negative ecological impacts mostly? If there is space for more words, would be useful to know if scientific literature is mostly discussing the negative impacts (and not possible positive ones, like hunting opportunities etc.)

RE: We added the perception part in the abstract in Lines 25-28 “Specifically, we compared news and scientific articles based on their emphasis: ecological, economic, and health impacts and the overall perception portrayed in the news: "positive" when the articles emphasized benefits from wild boar and "negative" when focused on damage and/or loss.”

- The abstract is a bit unclear – are you suggesting that a match between what scientists are concerned about needs to match that of news media coverage, because the news can drive management efforts of wild boar? E.g., if people want to hunt for boar and the government culls the population, people would be upset and not understand they’re negative? Please try to rephrase the abstract to be as targeted with your goal and implications.

RE: We do not think that interests should match between media coverage and scientific articles. However, a coincidence in interests could facilitate the management of invasive species. As we stated in the manuscript, the management, of invasive alien species may cause discontent and/or get disapproval from the media, which in turn can result in the interruption of management and control efforts. We have added the perception results and reworded the last part of the abstract to add clarity on this point (Lines 38-43): “Additionally, we found that 66.8% of news articles focused on “negative” aspects of wild boar, while 33.2% of news articles portrayed "positive" perceptions. This finding is very important because the management of invasive species such as wild boar usually requires lethal techniques, and the success of the programs depend on favorable social and political support. Good science communication is therefore key to helping scientists and managers perform more effective management actions. “

Introduction

- Line 60: please change “news or social media” to “news and/or social media”

RE: Done

- Line 61: What do you means by “increasing amounts of media”? People were interested in environmental issues regardless of media – do you mean that media coverage, especially with the growing massive use of social media, has been responsible for contributing to message sharing and awareness about environmental issues?

RE: No, we mean that in the recent decades there is an increasing number of media articles on environmental issues, and many studies have shown a positive correlation between media coverage and public awareness for the issue. For example:

Sampei, Yuki, and Midori Aoyagi-Usui. "Mass-media coverage, its influence on public awareness of climate-change issues, and implications for Japan’s national campaign to reduce greenhouse gas emissions." Global environmental change19.2 (2009): 203-212.

Mazur, Allan, and Jinling Lee. "Sounding the global alarm: Environmental issues in the US national news." Social studies of science 23.4 (1993): 681-720.

Both references were included in the article in line 59. 

- Lines 62-64: media coverage of a topic doesn’t always convinced people it’s true!

RE: True, but generally studies have shown media communication makes a positive contribution to understanding environmental problems. For example, see: Stamm, Keith R., Fiona Clark, and Paula Reynolds Eblacas. "Mass communication and public understanding of environmental problems: the case of global warming." Public understanding of science 9.3 (2000): 219.

- Lines 64-65: consider rephrasing this sentence to be more related to awareness e.g., “high visibility of species-specific media coverage can coincide with an increased level of perception and awareness of that species and its effects on the (local?) environment” – or something like that.

RE: We agree and modified the sentence. Lines 60-62: “Similarly, high visibility in media with respect to a species coincides with a high level of perception of that species and vice versa [9]. 

- Lines 65-67: this would entirely be based on the perspective from which that species is discussed, which you don’t mention – framing a species from a positive or negative perspective can result in the public perceiving a species as beneficial or harmful, respectively.

RE: Agree, we used the example on the mountain lions to explain it in Lines 62-65: “This phenomenon can potentially "improve" the image of a species, or conversely "demonize" it, turning public opinion to a negative perception. For example, media coverage of negative events, such as predator attacks (e.g. mountain lion on livestock), can amplify perceived risk and reduce support for conservation interventions [10].”

- Line 69: “conservation” needs to be followed up with terms like management, strategies, actions, etc.

RE: Added

- Line 78: although your article, thus far, appears to focus on convention news outlets, Bergman et al. 2022 (https://doi.org/10.1139/facets-2021-0112) highlight many examples of the effects of social media on wildlife conservation & invasive species management efforts from much more recent dates – would be useful to highlight this as social media is now an inherent part of news coverage for many people

RE: We appreciate the suggestion of Bergman et al 2022 citation. You are correct about the focus of our study on conventional news articles, nonetheless, we included the importance of social media on invasive species management in the discussion, Lines 55-57 “Media can play a key role in forming opinions by influencing people's interests, understanding, perception of a specific topic, and potentially action, through news and/or social media [1, 2, 3].” and Lines 261-273 “It is worth mentioning the increasing role of social media on invasive species research and management [3]. Social media, like other forms of communication, can be used effectively to share information about invasive species [51]. For example, Allain [52] found that the total number of a non-native freshwater turtle sightings within the UK recorded on Flickr from 2008 to 2018 was significantly greater than those submitted to Record Pool in the same period. Similarly, a crowdsourcing project on iNaturalist increased detection of wild boar (Sus scrofa) in Ontario, Canada [53]. Nonetheless, in the latter study the authors found that the reports of wild pig sightings were related to whether wild pigs were featured in media stories that directed the public on how to report sightings. This suggests that the number of reported wild pig sightings in Ontario was not a direct index of the abundance of wild pigs on the landscape, but rather an artifact of time of year and our media and outreach efforts. While the relationship of social media and invasive species was not the aim of this study, future research should look into how social media can improve invasive species communication, research and management.”

- Lines 80-81: I’m not sure what you mean here – media portrayals increased exposure? Not sure what these sentences mean.

RE: Changed to: “Previous studies have shown that the perception of invasive species can be influenced both positively or negatively by the way species are portrayed, via increased media exposure or by emphasizing specific points of view [17, 18, 19].” Lines 78-80

- Line 83: why did you use two different types of quotation marks here? This sentence also seems contradictory – it’s set up to discuss contentious issues, I believe you’re attempting to say however that media can explain reasons that culling efforts are required and that can smooth opposition? Unclear

RE: Changed to: “Similarly, invasive species management can become a contentious issue, as media can increase or decrease the discontent or social support of management strategies, which can subsequently determine the fate of a management plan (e.g. [18, 19]).” (Lines 80-83)

- Line 85: “social” and “media coverage” – is this supposed to be social media coverage? Or just news?

RE: Just news, changed to: “For example, media coverage (i.e. news) and protesters changed the management strategy of the hedgehog, an introduced species in Scotland, from trapping and euthanasia to a translocation initiative [18].” (Lines 83-85)

- Line 86: this example needs to be explained – don’t make us readers go searching for reference “13” – tell us what happened!

RE: Agree, as explained above.

- Line 87-89: did media reach policy makers? Was it not science? Or did they need the public to get on board with management efforts? Unfortunately these examples don’t support your work well because you fail to elaborate on them and make the connections clear. Please pick one example and delve into it a bit more. Same with lines 88-90.

RE: The media through news can undoubtedly influence policy makers since many times an environmental issue (e.g. the conservation of a species; or an environmental problem such as an oil spill) is initially installed in public opinion by the media. For this reason, the premise of our work is to highlight the importance of the media for the management of invasive alien species.

We removed one example and expanded details on two examples so make connections clearer. With these examples we try to show how the media, in this case through the news, can influence wildlife managers and policy making. 

The paragraph now reads (Lines 77-92): “Invasive species are a pervasive driver of global change with growing media representation (e.g. [14, 12, 15, 16]). Previous studies have shown that the perception of invasive species can be influenced both positively or negatively by the way species are portrayed, via increased media exposure or by emphasizing specific points of view [17, 18, 19]. Similarly, invasive species management can become a contentious issue, as media can increase or decrease the discontent or social support of management strategies, which can subsequently determine the fate of a management plan (e.g. [18, 19]). For example, media coverage (i.e. news) and protesters changed the management strategy of the hedgehog, an introduced species in Scotland, from trapping and euthanasia to a translocation initiative [18]. In addition, other studies have shown that the number of news articles and framing of invasive species impacts can influence legislator and public support of invasive species control and prevention [14]. For example, Miller et al. [12] found that congressional policy activity in the United States increased with media news, highlighting the negative impacts of wild boar on agriculture. Because news media can help establish parameters for public discourse, or how people think and talk about a subject [20], understanding the nature of media coverage may help scientists and managers perform more effective management activities [10] and/or identify research needs.”

- Line 91: consider changing “US” to “the United States.” Also, the newspapers themselves are not “negative” but I assume you mean that they highlight the negative impacts of invasive wild boars on agricultural practices. Please consider rewording to make this clear.

RE: As suggested above we deleted this example to improve clarity of the paragraph.

- Line 91-92: are these all the same wild pig species? You need to include species names please.

RE: Sus scrofa is the scientific name. "Wild boar" is the common name for the Eurasian morphotype; while "wild pig", "feral pig", "feral hog", “swine” etc. are generally used for the hybrid morphotypes between European wild boar and domestic pig (see: Long, J. L. 2003. Introduced mammals of the world: their history, distribution, and influence. CSIRO publishing). To avoid confusions, we used “wild boar” throughout the manuscript, and clarified the terminology in lines 98-99: “The wild boar (Sus scrofa), also known as wild pig, feral pig, swine, or hog [21] is one of the most widely distributed invasive species throughout the world [22]”

- Line 92-95: you haven’t provided any real clear evidence of this yet so this sentence falls a bit flat and doesn’t convince readers

RE: We believe that previous paragraphs and the bibliographical references support this statement.

- Line 96: the species name should be included at the first mention of that species

RE: Added.

- Lines 99-105: Consider rewriting this section to follow up each negative impact with an example e.g. you state they incur negative economic impacts, please then immediately follow up with your examples

RE: We edited this paragraph as suggested (Lines 93-106): “The wild boar (Sus scrofa), also known as wild pig, feral pig, swine or hog [21], is one of the most widely distributed invasive species throughout the world [22]. This species, native to Eurasia and North Africa, is considered one of the most harmful invasive alien species, endangering not only biodiversity but also the economy and human well-being [23, 22]. The wild boar is a generalist omnivore that feeds by overturning extensive areas of soil and vegetation. This disturbance causes many ecosystem-level effects by altering soil processes, reducing plant productivity, altering habitat availability and threatening biodiversity conservation [22, 24, 25, 26]. In addition, wild boar disturbance and predation has an economic impact on human productive systems as it is considered a crop pest, livestock predator, and competitor for forage and water resources, as well as a vector for several serious diseases that can affect wildlife, farms, domestic animals, and humans [22, 27, 28, 29]. Nonetheless, human perception of the wild boar is complex, because different stakeholders (e.g. agricultural producers, politicians, conservationists, hunters) perceive socio-economic values of this species differently (e.g. recreational hunting, tourism, cuisine, 30, 31]).”

- Line 106: what are the different stakeholder groups?

RE: We added the stakeholders as suggested. Lines 103-106: “Nonetheless, human perception of the wild boar is complex, because different stakeholders (e.g. agricultural producers, politicians, conservationists, hunters) perceive socio-economic values of this species differently (e.g. recreational hunting, tourism, cuisine, 30, 31]).”

- Line 108-109: You frame the purpose of your article to include a management aspect, but then don’t include that in this sentence. Also, are you exploring “public interest and news media” AND “scientific research” as two separate components? This is how it reads.

RE: We rewrote the paragraph to improve clarity (Lines 107-114): “Here, we explore the relationship between news media and scientific research on invasive species and how they can inform one another. We use the wild boar invasion in Argentina as a case study to conduct a content analysis in media coverage and scientific articles. Analysis of media coverage can provide insights into public opinion, stakeholder perspectives, and their influence on conservation actions [32]. We classified and compared news and scientific articles based on their emphasis: ecological, economic, and health impacts; as well as the overall perception (“positive” or “negative”), and discussed the implications for invasive species management.”

- Line 109: insert “the” before “wild boar invasion.” Is the invasion across the entire country? Or localized to certain areas?

RE: We added “the” as suggested. Currently wild boar is present in most of the country and their populations are growing and dispersing every year. 

Materials and Methods

- Lines 118-119: were all of these terms in parenthesis?

RE: No. The search in Google News is performed using each of the keywords (e.g. wild boar) defining the region (i.e. Argentina) using "site:AR"

- Line 121: how many articles in total did you review? How did you “sort” the results? By year or relevance?

RE: Initially 338 news were obtained with the keywords in Google News, which were then filtered down to 194 (see Results). We used the news that covers some aspect of the wild boar in Argentina. We included this information in the paragraph (Lines 120-125): “We filtered results to include only news articles by choosing the “News” tab in Google. We then reviewed each article of the 338 found to determine whether it was relevant to our search (i.e. news that directly covers some aspect of the wild boar in Argentina) and eliminated those that did not deal with wild boar or were redundant occurrences of the same article (i.e., the same news story replicated in several media). This search yielded 194 news items published from 2007 through 2020 (S1 Appendix).”

- Line 123: consider change the “–“ to (i.e., the same news story replicated in several media)

RE: Done. 

- Line 125: I believe you should indent this new paragraph

RE: Done.

- Lines 125-129: did you conduct a literature search in each of these databases separately using the exact same search terms? Did you use parenthesis? What language(s) did you include? Did you include review articles or only primary academic literature? How did you decide on those search terms? Why the year 1970?

RE: As stated in the manuscript for the news search in Google News we use "Sus scrofa" OR "jabalí" OR "chancho jabalí" OR "chancho salvaje" OR "chancho silvestre" OR "cerdo salvaje" OR "cerdo cimarrón" AND “site:ar”. The "site: AR" restricts the search only to Argentina. For scientific articles, we used "Sus scrofa" OR "wild boar" OR "wild pig" OR "feral pig" OR "jabalí" OR “chancho jabalí" OR "chancho salvaje" OR "chancho silvestre" OR "cerdo salvaje" OR "cerdo cimarrón", AND "Argentina". We use the terms in Spanish and English in the search (to cover the literature produced in both languages) and all types of articles were included, including reviews. The search terms were selected to cover all possible articles referring to the species. The limit of the years was eliminated because all the studies published for wild boar in Argentina in history were included.

- Line 135: this is inconsistent terminology – you used “ecological” before but now “ecologic”? What do you mean by culling – would this be management actions that are standardized and carries out to cull a large portion of a population but falls outside the class of consumption/hunting?

RE: We unified the term “ecological” throughout the manuscript. In this case we use the term “culling” refers deliberately reducing wildlife population for conservation purposes. We added this information in the manuscript for clarity (Lines 134-138) : “We classified each news and scientific article according to the main topic: 1) Economic (crop damage, livestock predation, sport hunting, poaching, animal husbandry, commercial exploitation, road accidents); 2) Ecological (culling – reduction of population for conservation purposes-, soil disturbance, predation, population growth); 3) Health (transmission of diseases to humans and other animals, attacks); and 4) others (e.g. sightings).”

- Line 136: consider changing “health to “public health”

RE: We refer to the concept “health” in general rather restricted to humans. This is related to the “One Health” concept (https://www.fao.org/one-health/) which is an integrated unifying approach that seeks to sustainably balance and optimize the health of people, animals and ecosystems. This approach recognizes that the health of people (public health), domestic and wild animals, plants and the general environment (including ecosystems) are closely related and interdependent.

- Line 128: how would wild boar be related to “animal husbandry”? Consider changing “when kept as pets” to “suitable as domestic house pets” to make that portion of the sentence more clear

RE: Sometimes the wild boar can be breed for hunting reserves or to produce meat. We changed to domestic pet as suggested (Lines 138-144): “We also recorded the perception of the news: "positive perception" meant the item emphasized benefits from wild boar such as sport hunting, commercial exploitation, husbandry (breeding for sport hunting or meat), and domestic pets, while we considered "negative perception" to be focus on impacts such as crop damage, livestock predation, diseases, culling, use of public areas, car collisions, ecological impact (e.g. soil damage, predation, competition for resources, disease transmission), population growth, and attacks.”

- Line 141: I believe you should change “topic and perception” to “topic and/or perception”

RE: Done.

- Line 142: this is known as meta-data, please as a comma after city

RE: Changed as suggested. 

- Line 144: how did you decide which geographic areas were “different”? Was it political or ecologically set?

RE: We adapted created the map based on the geographical classification of the Ministry of Environment and Sustainable Development of Argentina (https://www.argentina.gob.ar/ambiente). We included this information in map reference.

- Line 147: be careful with your wording – this was the first article that your search revealed (our search strategies are never perfect, you may have missed something!)

RE: Changed to: “The first news article on wild boar in Argentina according to our search was published in 2007, then the number grew exponentially starting in 2017, reaching a total of 194 articles to date (Fig 1, S1 Appendix).” (Lines 151-153)

- Line 151: ident the paragraph. You could refine wording and change to, “Topic frequency varied…”

RE: Changed as suggested

- Line 153: the % doesn’t make sense here – how could ecological impacts “dominate” when it was less than 1/3? Do publications = scientific articles? Please keep terminology the same throughout the article if this is what you mean. Please change “barely” to uncommon or a word more similar to that.

RE: The percentages sum less than 100 because some news did not fit any of the 3 major categories. We reworded the sentence to improve clarity as suggested (Line 156-157): “More than half of the news covered economic impacts of wild boar (56.1%), while ecological impacts reached 27% of the news, and health impacts almost 10% (Fig 2).”

- Line 156: can you give an example of ecological impacts? Perhaps a table that lists each of the categories within economic, ecological, and health would make it clearest for readers. Please add a comma after “finally”

RE: We added examples as suggested, it now reads (Lines 160-163): “Ecological issues in the news were related to population growth (11.4 %), culling (10.7 %), soil damage by rooting (3%) and competition or predation of native fauna (2%) (Fig 2). Finally, news on health focused on diseases (8 %), and attacks on humans (1.5 %) (Supporting Information II).”

- Lines 153-157: I don’t think your percentages are adding up, can you make clear which aspect falls under each category?

RE: We rewrote the paragraph to include information about categories and percentages, and fixed a bug in the value of sport hunting. Lines 163-170: “By contrast, the majority of the scientific articles focused on ecological aspects (64.6%), 20.8% focused on health impacts, only the 8.3% of scientific articles are related on economic impacts. Ecological impacts (41.7%) includes references mainly based on wild boar rooting impacts (35.5%), and a few articles covered topics on population growth (12.5%) and culling (10.4%) (Supporting Information II). Most of the scientific articles on health impacts focused on diseases (18.7%), and only one article included information about attacks on humans (2.1%). Finally scientific articles on economic impacts focused on animal husbandry (4.1%), and road accidents and poaching (2.1% each) (Fig 2).”

- Line 158: the 64.6% should be placed after “ecological aspects” and make this the end of a short sentence, starting the next sentence with your percentage information.

RE: Done as suggested, see above. 

- Line 159: “cover” should be “covered”

RE: Changed as suggested. 

- Line 162: again, please place a comma after “finally” – 8.3% of what? Did you miss words here?

RE: Changed as suggested, see response to line 153-157. 

- Line 164-165: I think it might be useful to redo this aspect of your article – it’s not that the articles focused on negative aspects of wild boar (although they are indeed negative consequences they incur) but instead almost the media was accurately portraying the invasive species as invasive (which inherently means negative/destructive/damaging/ etc) whereas “positive” instead treats invasive wild boar as naturalized and a species that can offer economic pros. This is an aspect I don’t believe you’ve really discussed and is critical.

RE: This is a good point, we added: “Interestingly we found that “negative” perceptions were associated with destructive, damaging activities by wild boar, while “positive” perceptions came from economic benefits. This finding is in line with those by Rodriguez-Rey et al. [40] in which the popularity index of invasive species was consistent with the public perceptions of their ecological and economic damage.” Lines: 220-224.

- Line 165-168: this is a method as you’ve already categorized and explained this

RE: Agree, deleted.

- Line 171: are the Santa Fe and Entre Rios the same area, or should you have added in a comma after Santa Fe?

RE: Different areas, we added a comma.

- Line 177: add a % after 40, and a comma after 30%

RE: Done

Discussion

- Line 175: I don’t think the word “emphasis” is appropriate, it seems more to be the actual topic of focus that there is a mismatch between the media and academic worlds

RE: We agree and change “emphasis” for “focus”.

- Line 176: remove the comma after “whereas”

RE: Done.

- Line 181: choose a different word at least once instead of writing generally and general twice

RE: We removed the word “general”.

- Line 182: why is this the first time you have a written-out citation? Is this a mistake?

RE: We deleted “Gozlan et al” and leave the number reference [4]

- Line 183: I don’t believe this is striking…in fact, I’d say this is very common worldwide and what I’d expect

RE: We agree and remove “striking”.

- Line 189-192: this is a long, somewhat confusing, sentence

RE: We rewrote the sentence, lines 194-197: “Indeed, several studies show agriculture or plantation losses due to wild boar disturbance in the United States (e.g., [27, 33, 34, 35], but the lack of scientific information on economic losses plus difficulties in comparing different studies preclude the estimation of the annual economic impact [36].

- Line 192-193: how? Big statement but I don’t clearly see the connection…

RE: We agree and rewrote the sentence including an example. Lines 197-200: “This lack of information may also limit and/or hinder legislation and the design of management strategies. For example, the limitation or lack of knowledge of some aspects of invasive alien species has been recognized as one of the main threats to dealing with the management of these species in Europe [37].”

- Line 195: this sentence needs to be rephrased to specifically describe that it was related to boar and not broadly all invasive species management

RE: Changed to: “For example, Miller et al. [12] found that media coverage in the United States determines wild boar policy, as news on negative impacts resulted in an increased congressional policy activity and development.” Lines 201-204.

- Line 199: please change to “Invasive species legislation in the Pampas region allows for an extended hunting season and increase (??) area to control damages wild boar can incur to agricultural areas” or something like that to be more clear

RE: We rewrote the sentence to be more clear, it now reads: “In Argentina, wild boar was declared a crop and livestock pest in the Pampas region where conflicts with agriculture arise (e.g. Decree 279. Resolution 272 of the Ministry of Health, Province of Buenos Aires, 2019). Wild boar legislation in this region allows year-round hunting to reduce wild boar damage in agricultural lands.” Lines 204-207.

- Line 200: please put the United States before those bills instead of at the end

RE: Done.

- Line 206: please rephrase to, “; our results suggest there is considerable media attention focused on wild boar, however the aspects described often relate to social outcomes” or something

RE: Changed to: “Previous studies have shown that non-native species receive little media coverage [9, 16]; however, our results suggest there is considerable media attention on wild boar, but the aspects described often relate to social outcomes.” Lines 213-216.

- Line 208: suggest needs to be plural; change “need to” to “must” and I would argue that this needs to be done in both conventional news outlets and also social media

RE: Agree, we rewrote the sentence, lines 216-218: “This finding also suggests that scientists must better communicate their knowledge to the public. Science communication is needed not only for increasing scientific literacy but also to avoid management conflicts [10, 18, 14] and ensure that management is evidenced-based [38].”

- Line 210: not only to avoid management conflicts, but ensure that management is evidence-based and can support healthy wildlife populations and nature conservation

RE: We agree with the suggestion, and rewrote the sentence. Lines 217-218: “Science communication is needed not only for increasing scientific literacy but also to avoid management conflicts [10, 18, 14] and ensure that management is evidenced-based [38].”

- Line 213-216: this is a long sentence and should be divided to give ample time to discussion about ecotourism and charismatic species (I’ll note however that this doesn’t seem to relevant to your article…)

RE: We believe that it is a good example of an invasive charismatic species, involving in a complex socio-ecological context. We split the sentence according to the reviewer's suggestion: “For example, lethal control initiatives of a population of feral hippopotamuses (Hippopotamus amphibius) in Colombia were abandoned due to strong public opposition [19]. Hippopotamuses are a charismatic species and are valued by local communities because they attract tourists to the area [19].” 

Lines 225-228.

- Line 222: inducement? This word doesn’t make sense here

RE: We agree and change the concept for “key element”. Line 236.

- Line 224-225: you just said earlier though that the US is so different?

RE: We don’t understand this comment. We are stating that our results from wild boar invasion in Argentina should be similar world-wide.

- Line 228: add “that” after “parasites”

RE: Added.

- Line 229: Scientists traced the genetic lineage of H1N1 swine flu to a strain that emerged in 1998 in U.S. factory farms…swine flu is also emerging from wild boar? I didn’t know this…am I misinterpreting? The article you cite I believe refers specifically to ASF

RE: It is our mistake. We meant the African Swine Fever and not to the swine flu. The sentence was modified. Lines 241-244.

- Line 230-232: consider selecting a word other than “probably” for this sentence and make the link clearer here between increased awareness about zoonotic diseases

RE: Changed to: “However, health impacts of wild boar will likely become a more conspicuous topic both in news and scientific articles after the covid-19 pandemic and consequently increased awareness of the importance of zoonotic diseases [48].” Lines 244-246.

- Line 234: did you list a statistical statistic for this finding? I may have missed that you found no significant difference between geographic areas and sources of news and research articles

RE: There is no statistics, we mean that news and scientific articles originated in the same regions where wild boar is currently present. We rewrote the sentence: “The news and research articles originated in the same regions where wild boar is distributed in Argentina.” Lines 247-248.

- Line 234: remove the comma after Patagonia

RE: Done.

- Line 236: put “the Pampas region” in parenthesis if this area is in central Argentina

RE: Done.

- Line 238: production of an agricultural crop?

RE: Added as suggested. 

Figures

Love the map figure

RE: Thanks!

---

## [Decision Letter · Decision Letter 1]

12 Dec 2022

Mismatch between media coverage and research on invasive species: the case of wild boar (Sus scrofa) in Argentina

PONE-D-22-17828R1

Dear Dr. Ballari,

We’re pleased to inform you that your manuscript has been judged scientifically suitable for publication and will be formally accepted for publication once it meets all outstanding technical requirements.

Kind regards,

Jorge Ramón López-Olvera

Academic Editor

PLOS ONE

Additional Editor Comments (optional):

Dear Authors,

I received the comments from the reviewer #2 who assessed the first version of your manuscript some days ago, who found that all her/his comments had been adequately addressed and now the manuscript is ready for acceptance. I apologize for the time spent since I received this second report from Reviewer #2, while I was waiting for the reviewer #1 of the first revision to check whether she/he considered the comments satisfactorily addressed also. Unfortunately, reviewer #1 did not answer to two consecutive calls to assess the revised version of your manuscript, so I undertook myself the task of assessing it. Reviewer#1 proposed really minor changes that make unworthy further delay in your publication.

It is therefore my pleasure to inform you that, according to the comments by reviewer #2 and my own assessment, your revised submission can be now considered suitable for publication in PLoS ONE.

Best regards,

Reviewers' comments:

Reviewer's Responses to Questions

**Comments to the Author**

1. If the authors have adequately addressed your comments raised in a previous round of review and you feel that this manuscript is now acceptable for publication, you may indicate that here to bypass the “Comments to the Author” section, enter your conflict of interest statement in the “Confidential to Editor” section, and submit your "Accept" recommendation.

Reviewer #2: All comments have been addressed

2. Is the manuscript technically sound, and do the data support the conclusions?

Reviewer #2: Yes

3. Has the statistical analysis been performed appropriately and rigorously? 

Reviewer #2: N/A

4. Have the authors made all data underlying the findings in their manuscript fully available?

Reviewer #2: Yes

5. Is the manuscript presented in an intelligible fashion and written in standard English?

Reviewer #2: Yes

6. Review Comments to the Author

Reviewer #2: (No Response)

7. PLOS authors have the option to publish the peer review history of their article (what does this mean?). If published, this will include your full peer review and any attached files.

Reviewer #2: No

---

## [Editor Report · Acceptance letter]

15 Dec 2022

PONE-D-22-17828R1 

Mismatch between media coverage and research on invasive species: the case of wild boar (*Sus scrofa*) in Argentina 

Dear Dr. Ballari:

I'm pleased to inform you that your manuscript has been deemed suitable for publication in PLOS ONE. Congratulations! Your manuscript is now with our production department. 

Kind regards, 

on behalf of

Dr. Jorge Ramón López-Olvera 

Academic Editor

PLOS ONE